# Broad-Based Influenza-Specific CD8^+^ T Cell Response without the Typical Immunodominance Hierarchy and Its Potential Implication

**DOI:** 10.3390/v13061080

**Published:** 2021-06-05

**Authors:** Miaojuan Huang, Rong Xu, Cristina Triffon, Nicole Mifsud, Weisan Chen

**Affiliations:** 1Department of Biochemistry and Genetics, La Trobe Institute for Molecular Science, La Trobe University, Bundoora, VIC 3086, Australia; 18212145@students.latrobe.edu.au (M.H.); C.Triffon@latrobe.edu.au (C.T.); 2Nanobiotechnology Laboratory, Australian Centre for Blood Diseases, Monash University/Alfred Health, Melbourne, VIC 3004, Australia; r.xu@latrobe.edu.au; 3Infection and Immunity Program, Department of Biochemistry and Molecular Biology, Biomedicine Discovery Institute, Monash University, Clayton, VIC 3168, Australia; nicole.mifsud@monash.edu

**Keywords:** influenza A virus, HLA, antigen presentation, CD8^+^ T cell epitope, immunodominance

## Abstract

Syngeneic murine systems have pre-fixed MHC, making them an imperfect model for investigating the impact of MHC polymorphism on immunodominance in influenza A virus (IAV) infections. To date, there are few studies focusing on MHC allelic differences and its impact on immunodominance even though it is well documented that an individual’s HLA plays a significant role in determining immunodominance hierarchy. Here, we describe a broad-based CD8^+^ T cell response in a healthy individual to IAV infection rather than a typical immunodominance hierarchy. We used a systematic antigen screen approach combined with epitope prediction to study such a broad CD8^+^ T cell response to IAV infection. We show CD8^+^ T cell responses to nine IAV proteins and identify their minimal epitope sequences. These epitopes are restricted to HLA-B*44:03, HLA-A*24:02 and HLA-A*33:03 and seven out of the nine epitopes are novel (NP_319–330_^#^ (known and demonstrated minimal epitope positions are subscripted; otherwise, amino acid positions are shown as normal text (for example NP 319–330 or NP 313–330)), M1_124–134_, M2_7–15_, NA_337–346_, PB2_39–49_, HA_445–453_ and NS1_195–203_). Additionally, most of these novel epitopes are highly conserved among H1N1 and H3N2 strains that circulated in Australia and other parts of the world.

## 1. Introduction

The 1918 Spanish influenza pandemic resulted in over 50 million deaths worldwide and remains one of the deadliest plagues ever experienced in human history [1]. Annually, influenza infection continues to cause epidemics and is responsible for 3–5 million severe infections and 500,000 deaths worldwide https://www.who.int/en/news-room/fact-sheets/detail/influenza-(seasonal) (accessed on 27 April 2021), even though influenza vaccines are widely available. These vaccines are designed to elicit neutralising antibodies, however, their efficacy in elderly and young are still debated [2]. Importantly, CD8^+^ T cells play an essential role in controlling influenza infections [3,4], as they mainly recognise antigens derived from more conserved internal viral proteins, such as NP [5,6] and memory influenza-specific CD8^+^ T cells are associated with enhanced recovery from novel influenza infection [7].

CD8^+^ T cells recognise viral peptides bound to HLA-I molecules presented on the surface of infected cells [8]. Many peptides generated from influenza proteins could potentially be presented to CD8^+^ T cells, however, only a small fraction of such peptides are recognised by virus specific CD8^+^ T cells with a measurable response. Among these, often only one or two peptides are able to stimulate a large response, a phenomenon known as immunodominance [9,10]. Most studies that investigated the underlying mechanisms of immunodominance often used syngeneic mouse models, such as the well-studied influenza A virus (IAV) infection model in C57BL/6 and BALB/c mice [11,12]. Although such models are well suited for studying factors such as peptide-binding ability to MHC, antigen-processing efficiency, and TCR repertoires and their impact on immunodominance [11,12,13,14,15], they are not the best means to study the impact of MHC polymorphism on immunodominance. Boon et al. studied CD8^+^ T cell responses in individuals with single HLA-A or -B allele mismatch and clearly demonstrated that some HLA alleles play bigger roles in presenting IAV antigens [16,17]. Similarly, using HLA-A2 positive donors and a systematic CD8^+^ T cell assessment approach, our laboratory was able to show the immunodominant CD8^+^ T cell response to HLA-A2/M_58–66_ could clearly be affected by other HLA alleles [5]. If such observation is applicable to all HLA-A and -B allele combinations, it is then possible that some individuals may never present immunodominant CD8^+^ T cell response simply because their HLA alleles do not select such immunodominant epitopes. As a direct result, the CD8^+^ T cell responses in these individuals will likely be broad and have many subdominant determinants without a typical immunodominance hierarchy; this type of response has not been previously reported for CD8^+^ T cells specific to IAV infections, although our laboratory has previously reported such an IAV-specific CD4^+^ T cell response [18].

In this study, we report a broad-based IAV-specific CD8^+^ T cell response from a healthy donor to nine IAV antigens. Interestingly, the CD8^+^ T cell responses specific to the epitopes from the nine IAV proteins showed similar intensity. We believe that an in-depth understanding of such CD8^+^ T cell responses could help vaccine design and may predict severe clinical outcome.

## 2. Materials and Methods

### 2.1. Human Peripheral Blood Mononuclear Cells (PBMCs) and Antigen Presenting Cells (APCs)

PBMCs of healthy donors used in this study were obtained from the Australian Red Cross Blood Service (ARCBS) in Melbourne under the agreement 12-07VIC-17. PBMCs were isolated from blood buffy coats using Ficoll-Hypaque gradient centrifugation, then resuspended in FCS (Hyclone, South Logan, UT, USA) containing 10% dimethyl sulfoxide (DMSO) and stored in liquid nitrogen until use. The autologous B lymphocyte cell line (BLCL) was made from the donor’s PBMCs using EBV transformation [5], and cultured in RPMI 1640 supplemented with 10% FCS (Hyclone) (RF-10).

### 2.2. Culture Medium, Synthetic Peptides, Antibodies and Viruses

Unless otherwise stated, antigen presenting cells were cultured in RF-10 containing 50 µM 2-ME, 2 mM L-glutamine and antibiotics (penicillin 100 U/mL and streptomycin 100 µg/mL) (all from Gibco, Grand Island, NE, USA)). Peptides used in this study were synthesized by Mimotopes (Clayton, VIC, Australia). Peptides were reconstituted to a concentration of 1 mM in DMSO. APC-labelled anti-CD8 was purchased from BD Biosciences, and PE-cy7 labelled anti-IFNγ was purchased from Invitrogen (San Diego, CA, USA).

The IAV strain A/Puerto Rico/8/1934 H1N1 (PR8) was propagated in 10-day old embryonated chicken eggs. Two days after infection, the allantoic fluids were collected, aliquoted and stored at −80 °C in the freezer until use. Recombinant vaccinia viruses (rVV) encoding individual IAV proteins HA, NA, NP, NS1, NS2, M1, M2, PB1, PB2, PB1F2, PA, and the empty vector rVV-CR19 were kind gifts from Drs Jonathan Yewdell and Jack Bennink (National Institutes of Health, Bethesda, MD, USA).

### 2.3. Generation of Poly-Specificity CD8^+^ T Cell Lines In Vitro

For establishing primary antigen-specific CD8^+^ T cell lines, 1 × 10^6^ PBMCs were resuspended in 200 µL acidic medium and infected with PR8 at an MOI (Multiplicity of Infection) of 10 for 1 h in a 37 °C water-bath. Next, 2 mL of RF-10 was added to the cells and incubated for a further 4 h in a 37 °C 5% CO_2_ incubator. The infected PBMCs served as APCs and subsequently co-cultured with 9 × 10^6^ PMBCs (responders) for two weeks in T cell culture medium containing 20 IU/mL of hrIL-2.

For CD8^+^ T cell re-stimulation, BLCLs were used as stimulators; they were infected with PR8 overnight (16 h), then irradiated for 100 Grey before being added to T cells in a T cell culture medium containing 40 IU/mL of hrIL-2. The ratio of T cells to APCs was generally 10:1.

### 2.4. rVV Infection for Identifying Protein-Specific CD8^+^ T Cells

APCs were infected with a panel of individual IAV protein encoding rVVs (rVV-HA, rVV-NA, rVV-NP, rVV-NS1, rVV-NS2, rVV-M1, rVV-M2, rVV-PB1, rVV-PB2, rVV-PB1F2, rVV-PA) or the empty vector rVV-CR19. Briefly, APCs were washed with 0.1% BSA/PBS twice, then re-suspended in the same buffer and infected with rVV encoding individual IAV proteins at MOI of 10. Cells were then incubated in a 37 °C water-bath for 1 h and were gently vortexed every 15 min. After the first hour of infection, cells were topped up with 2 mL fresh RF-10 and incubated in a 37 °C 5% CO_2_ incubator for a further 16 h.

For identifying a specific antigenic protein, IAV-specific CD8^+^ T cells were screened with the above rVV infected APCs.

### 2.5. Identification of Novel IAV-Specific CD8^+^ T Cell Epitopes and Its Minimal Epitopes

Once the specific antigenic proteins were identified, polyclonal CD8^+^ T cells were directly used to work out the specific peptide regions by screening a whole set of synthetic 18mer overlapping peptides with 6 amino acid shifts. For identifying the minimal epitopes, the shorter peptides within the 18mer region of interest were predicted according to the peptide-binding motifs and screened by single peptide-specific CD8^+^ T cell lines.

### 2.6. HLA Restriction Assay

BLCLs with partial donor HLA match were pulsed with 10^−6^ M of relevant peptides for 1 h at room temperature. These cells were then extensively washed to remove free peptides and were used as APCs in a co-culture with antigen-specific CD8^+^ T cells for 4 h in the presence of Brefeldin A (BFA). HLA-restriction was assessed by intracellular cytokine staining (ICS) for IFN-γ production.

### 2.7. Intracellular Cytokine Staining (ICS) and Antigen Presentation Kinetics Assay

The IFN-γ ICS assay was performed as previously described [5]. For antigen presentation kinetics assay, the detailed method was previously published [19]. Briefly, the antigen-presentation of PR8/rVV-infected autologous BLCLs and homozygous BLCLs were stopped by the addition of BFA (Sigma-Aldrich, St Louis, MO, USA) at different time points. Antigen-specific CD8^+^ T cells were then added for 4 h to assess the antigen-presentation at that particular time point; the T cells were then harvested and synchronized on ice before being stained together in a standard ICS. Samples were analysed using a BD FACS Canto II machine (BD Biosciences, Franklin Lakes, NJ, USA) and FlowJo software (Tree Star Inc, Ashland, NE, USA).

## 3. Results

### 3.1. A Systematic Approach for Identifying IAV-Specific CD8^+^ T Cell Responses

A two-step systematic approach [5] was used to (i) identify the viral proteins that stimulated CD8^+^ T cell responses (Figure 1a), and then to, (ii) determine minimum antigenic peptide sequences and their HLA restrictions (Figure 1b). First, IAV-specific polyclonal CD8^+^ T cell lines were established by infecting 10% of autologous PBMCs with PR8 to stimulate all potential IAV-specific memory CD8^+^ T cells among these PBMCs. After two weeks, CD8^+^ T cells specificities were assessed by infected autologous BLCLs with a panel of rVVs encoding 11 individual IAV proteins followed by synthetic overlapping 18mer peptides via ICS for IFN-γ production. Once the specific antigenic regions were identified, the minimal epitope sequences were determined using either overlapping 13mer peptides and/or shorter predicted peptides within the active 18mer sequence. Partially HLA-matched cell lines were used to determine these epitopes’ HLA-restrictions.

### 3.2. Broad-Based IAV-Specific CD8^+^ T Cell Responses to Multiple IAV Proteins

To identify a donor with broad-based CD8^+^ T cell responses to IAV antigens, polyclonal IAV-specific CD8^+^ T cell cultures from PBMCs of 8 HLA-A2^+^ healthy donors and 7 HLA-A2- healthy donors [5,6] and 6 other donors (no HLA typing, Appendix A) were established as described in Figure 1a and were used to screen rVV encoding individual IAV proteins. Only one of the cultures from the HLA-A2^−^ healthy donors (donor 4 [6]) showed broad CD8^+^ T cell responses to nine IAV proteins (PA, NA, HA, M1, M2, NP, NS1, PB1, and PB2) but did not to PB1F2 and NS2 (Figure 2). Although NP was previously demonstrated to be the most dominant antigen in most healthy donors [5,6], that is not the case in this individual as most of the CD8^+^ T cell responses specific to the nine IAV proteins showed similar intensity. Each rVV infected the cell line with similar efficiency, demonstrated by their similar rVV antigen expression levels (Appendix A).

### 3.3. Characterising Antigenic Peptide Regions in Multiple IAV Proteins

Following the identification of the IAV proteins responded to by these CD8^+^ T cells, a whole set of synthetic 18mer overlapping peptides covering the entire length of these proteins were screened by the above bulk T cell line to further identify antigenic regions in each protein (Appendix A). The polyclonal CD8^+^ T cell line was subsequently enriched by CD3CD8 downregulation-guided sorting, using APCs infected by a single rVV encoding individual IAV protein to activate the specific CD8^+^ T cells [20]. The enriched CD8^+^ T cell lines were then tested with the previously used whole set of synthetic 18mer peptides to further confirm the results obtained from the bulk T cell line used in Figure 2. As shown in Figure 3 (results derived from the enriched CD8^+^ T cell lines), the antigenic regions from the above-mentioned nine IAV proteins were: M2 7–24, M1 121–138, NP 313–330, NS1 193–210, PB1 493–510, PB2 37–54, HA 439–456, NA 331–348, and PA 121–138. Interestingly, only one antigenic region in each IAV protein was recognised by each enriched specific CD8^+^ T cell line except for NS1, in which we failed to detect a clear antigenic peptide region although >10% T cells responded to rVV-NS1.

### 3.4. Minimal Epitopes of Differing Lengths Are Presented by Three HLA Alleles

It has been demonstrated that MHC is one of the critical determining factors of immunodominance [21,22] and the MHC-I peptide-binding groove can only accommodate peptides generally of 8–12 amino acid in length [23,24]. Therefore, minimal epitope sequences and their HLA restrictions were investigated to better understand such a broad-based CD8^+^ T cell response. First, single peptide-specific CD8^+^ T cell lines were generated either by 18mer peptide stimulation as shown in Figure 1b or by enriching through CD3CD8 downregulation-guided sorting [20]. All the newly identified 18mer peptides were shown to be restricted to HLA-A*33:03, HLA-A*24:02, or HLA-B*44:03 (Appendix A). Interestingly, not a single peptide was shown to be restricted to HLA-B*46:01 expressed also by this donor. All three HLA molecules have stringent peptide-binding motifs [25,26,27]; for example, HLA-B*44 molecules require a glutamate (E) in position two (P2), and at the C-terminus either a tyrosine (Y) or phenylalanine (F), and less often tryptophan (W) [25]. Using this information, shorter peptides within the 18mer region of interest were predicted according to the peptide-binding motifs and screened by single peptide-specific CD8^+^ T cell lines.

### 3.5. Identifying the Core Sequences of the Epitopes Presented by HLA-B*44:03

For HA-specific CD8^+^ T cells, HA 439–456 (AELLVL**LENERTLDFHDS**) and HA 445–462 (**LENERTLDFHDS**NVKNLY) peptides (overlapping region shown in bold) revealed a positive CD8^+^ T cell response among the overlapping 18mer peptides (Figure 3). Moreover, these two neighbouring peptides stimulated T cell responses to a similar degree, indicating the minimal epitope was likely within the shared sequence HA 445–456 (**LENERTLDFHDS**). We then worked out these two positive peptides were restricted to HLA-B*44:03 (Appendix A) using donor HLA-matched APCs from Table 1. As HLA-B*44:03 requires P2 and C-terminal anchor residues E and Y/F respectively, HA 445–453 **LENERTLDF** was predicted as the minimal epitope. The peptide and its single amino acid truncated versions were synthesised and tested in a peptide titration experiment. As shown in Figure 4a-i, HA 445–453 stimulated a CD8^+^ T cell response even at a concentration of 10^−9^ M. However, HA 446–453 was only able to stimulate a weaker T cell response at much higher peptide concentration (10^−6^ M), while HA 445–452 lost all the activity. Therefore HA_445–453_ is the minimal epitope; subsequently its HLA-B*44:03 restriction was again confirmed (Figure 4a-ii). According to the Immune Epitope Database (IEDB) (www.immuneepitope.org) (last accessed on 27 April 2021), HLA-B*44:03 restricted HA_445–453_ is a novel epitope identified in this study.

Similarly, the M2 specific CD8^+^ T cells recognised both M2 1–18 (MSLLTE**VETPIRNEWGCR**) and M2 7–24 (**VETPIRNEWGCR**CNGSSD) to a similar degree (Figure 3) and showed HLA-B*44:03 restriction (Appendix A). M2 7–15 (**VETPIRNEW**) as was predicted as the minimal epitope within the shared M2 7–18 (**VETPIRNEWGCR**) region. Peptide titration experiments were performed for M2 7–15, M2 8–15 and M2 7–14 and very similar results were obtained (Figure 4b-i) as those acquired above, and HLA-B*44:03 restriction was confirmed (Figure 4b-ii). According to the IEDB, M2_7–15_ is a novel epitope identified in this study.

In similar fashion, NP-specific CD8^+^ T cells recognised 18mer peptides NP 313–330 (YSLIRP**NENPAHKSQLVW**) and NP 319–336 (**NENPAHKSQLVW**MACHSA) (Figure 3) and were restricted by HLA-B*44:03 (Appendix A). NP 319–330 (**NENPAHKSQLVW**) was predicted as the minimal epitope, which was confirmed by the decreased and total loss of T cell activating capacity of the amino- and carboxyl-terminal single amino acid truncation peptides, respectively (Figure 4c-i). This is a longer minimal epitope containing 12 amino acids and again HLA-B*44:03 restriction was confirmed (Figure 4c-ii). Thus, NP_319–330_ is the minimal epitope and is also the longest HLA-B*44:03-restricted novel IAV epitope according to current information available on IEDB.

The PB2-specific CD8^+^ T cells (HLA-B*44:03 restricted, Appendix A) showed an interesting pattern of peptide recognition as they were slightly activated by PB2 31–48 (IKKYTSGR**QEKNPALRMK**) but well activated by PB2 37–54 (GR**QEKNPALRMKW**MMAMK) (Figure 3), indicating PB2 37–54 must contain the minimal peptide sequence and PB2 31–48 likely misses a single C-terminal amino acid. PB2 39–49 (**QEKNPALRMKW**) was therefore predicted as the minimal epitope. This was confirmed by the loss of T cell activating capacity of the two peptides with a single amino acid truncation at either end of PB2 39–49 (Figure 4d-i). The restricting HLA for PB2_39–49_ was subsequently reconfirmed as HLA-B*44:03 (Figure 4d-ii). Again, this is a novel epitope by comparing with those listed in the IEDB.

The NS1-specific CD8^+^ T cells, although recognised rVV-NS1 infected APC (15% produced IFN-γ), they basically failed to recognise any NS1 18mers, except for very limited T cell activation induced by NS1 187–204 (WNDNTV**RVSETLQRFAWR**) and NS1 193–210 (**RVSETLQRFAWR**SSNENG) (<1%) (Figure 3) and was restricted to HLA-B*44:03 (Appendix A). Although the NS1 18mer responses were very low, these CD8^+^ T cells were specifically enriched using NS1 193–210 pulsed APCs and CD3CD8 downregulation-guided sorting. NS1 195–203 (**SETLQRFAW**) was predicted as the minimal epitope. Indeed, T cell activation was significantly reduced when they were stimulated by peptides with a single amino acid truncation at either end of NS1 195–203 (Figure 4e-i). Therefore, NS1_195–203_ is the minimal epitope restricted to HLA-B*44:03 (Figure 4e-ii). Again, comparing to those listed in the IEDB, this epitope is also novel. We believe that the bulk of NS1-specific CD8^+^ T cells might recognise a T cell epitope(s) either encoded by an alternative reading frame or derived from spliced peptides [28,29].

### 3.6. Identifying the Core Sequences of the Epitopes Presented by HLA-A*33:03

Similar to HLA-B*44:03, HLA-A*33:03 also possesses relatively stringent peptide-binding motifs. Although it tolerates alanine (A), isoleucine (I), leucine (L), phenylalanine (F), tyrosine (Y) or valine (V) at P2, it requires almost exclusively an arginine (R) at the C-terminus [26].

The M1-specific CD8^+^ T cells responded to M1 121–138 (AGA**LASCMGLIYNR**MGAV) and M1 127–144 (**CMGLIYNR**MGAVTTEVAF) 18mers (Figure 3) and the response was restricted by HLA-A*33:03 (Appendix A). As the two 18mer peptides only contained a single arginine (R) at position 134, M1 127–144 still had some T cell activation capacity (Figure 3) likely due to weak stimulation at high concentration. The P2 residue was predicted as 125 A at the amino-terminus rather than 124 L or 123 A because 12mer and 13mer epitopes are rare, therefore, M1 124–134 (**LASCMGLIYNR**) and its single terminal truncation peptides were synthesised. As P4 is a cysteine, to avoid cysteinylation and peptide dimerization, these peptides were assessed in the presence of 500 µM TCEP [30]. As shown in Figure 4f-i, both M1 124–134 and M1 125–134 were quite potent in the titration experiment, while M1 124–133 was no longer capable of T cell activation due to missing the C-terminal anchor residue arginine. Judging from the titration curves, M1 124–134 is more potent than M1 125–134, therefore, M1_124–134_ is the minimal epitope, restricted to HLA-A*33:03 (Figure 4f-ii), and again this is a novel epitope.

The NA-specific CD8^+^ T cells responded to two neighbouring 18mer peptides NA 331–348 (VKGFSY**RYGNGVWIGRTK**) and NA 337–354 (**RYGNGVWIGRTK**SHSSRH) (Figure 3) indicating the minimal epitope should be within the shared region NA 337–348 (**RYGNGVWIGRTK**). NA 337–346 (**RYGNGVWIGR**) was therefore predicted as the minimal epitope. This was confirmed by peptide titration experiments showing reduced T cell activation capacity of the two peptides with a single amino acid truncation at either end of NA 337–346 (Figure 4g-i). NA_337–346_ was confirmed to be HLA-A*33:03 restricted (Figure 4g-ii) and it is not yet in IEDB.

### 3.7. Identifying the Core Sequences of the Epitopes Presented by HLA-A*24:02

HLA-A*24:02 peptide-binding motifs prefer a Tyrosine (Y) at P2 and a phenylalanine (F) or leucine (L) at the C-terminus [27]. According to PB1 18mer mapping result (Figure 3), PB1 specific CD8^+^ T cells responded to PB1 493–510 (TSFFY**RYGFVANFSMELP**) only and the response was restricted by HLA-A*24:02 (Appendix A). Therefore, the minimal epitope was considered to be within PB1 498–510 (**RYGFVANFSMELP**). As the HLA-A*24:02 C-terminus anchor can be either leucine (L) or phenylalanine (F), it was then possible that either PB1 498–505 (**RYGFVANF**) or PB1 498–509 (**RYGFVANFSMEL**) could be the minimal epitope. As PB1 498–505 has been published [31], PB1 498–509 and the single amino acid truncated at either end PB1 499–509, PB1 498–508 were synthesised. To avoid unexpected peptide trimming by serum proteases [32], peptide titrations were conducted in the presence and absence of FCS. As shown in Figure 4h-i (in the presence of FCS, solid lines), PB1 498–508 and 498–509 could be titrated to 10^−9^ M; their stimulating capacity was the same as that of the previously published PB1_498–505_ (Figure 4h-i). However, in the absence of FCS (dotted lines), PB1_498–505_ was 10-fold more potent than PB1 498–508 and PB1 498–509, likely indicating that the longer peptides were processed into PB1_498–505_ by FCS-derived proteases. Therefore, we conclude that the minimal epitope is the published PB1_498–505_ restricted to HLA-A*24:02 (Figure 4h-ii).

The PA-specific CD8^+^ T cells recognised two neighbouring overlapping 18mer peptides PA 121–138 (GVTRRE**VHIYYLEKANKI**) and PA 127–144 (**VHIYYLEKANKI**KSEKTH) to the same degree (Figure 3), indicating that the minimal epitope is located in the shared sequence PA 127–138 (**VHIYYLEKANKI**). PA_130–138_ (**YYLEKANKI**) was predicted as the minimal epitope and the response was further confirmed by IAV-specific CD8^+^ T cells, however, this HLA-A*24:02-restricted epitope was previously identified [33].

### 3.8. Epitope Conservancy Among All the IAV Strains Circulated in Australia during 1971–2021 Period

To analyse the conservancy of all the novel immunogenic epitopes discovered in this study, IAV protein sequences were aligned, and amino acid differences were scored for the IAV strains circulated in Australia during the 50 years between 1971 and 2021. According to our analysis, two out of the seven novel IAV epitopes are 100% conserved in these viruses, including NP_319–330_, M1_124–134_ in both H1N1 and H3N2. For M2_7–15_, NA_337–346_, PB2_39–49_ and NS1_195–203_, although amino acid differences are found, the HLA anchor residues of these three epitopes are still conserved (Table 2). The information of circulated IAV strains (Table 3) was obtained from the National Center for Biotechnology Information (NCBI) influenza virus resource database accessed on 13 April 2021. Based on the above analysis, it is highly likely that most IAV-specific CD8^+^ T cells studied here would be able to fully or partially recognise cells infected by these different IAV strains including the 2009 H1N1 pandemic strain. Taken together, our results revealed multiple novel CD8^+^ T cell epitopes restricted to the individual’s HLA-A and B molecules with 100% conserved epitope sequences among IAV strains.

### 3.9. Quantitative Assessment of HLA-I Peptide Presentation to the Antigen-Specific CD8^+^ T Cells

It has been demonstrated that the speed of antigen processing and presentation influences the hierarchy of the CD8^+^ T cell response during viral infections [11,34,35,36]. To further understand the basis of such a broad-based IAV-specific CD8^+^ T cell response observed in this study, the antigen presentation kinetics of the above T cell epitopes were investigated using a BFA kinetics assay as slow antigen presentation could potentially explain the observed overall small and lack of immunodominant responses in this individual. BFA blocks pMHC-I egressing the ER [37] and therefore “freezes” pMHC-I complexes on the surface of virus infected APC at the time of addition and allows the assessment of pMHC-I antigen presentation level at any time point after infection [19]. This method also allows the comparison of different T cell populations engaging the same APC which presents various CD8^+^ T cell epitopes simultaneously. Additionally, the presentation kinetics of a well-studied immunodominant epitope M1_58–66_ was used as the “golden standard”.

The presentation kinetics results are displayed by their epitopes’ HLA restriction (HLA-B*44:03, HLA-A*24:02 and *33:03). For the HLA-A*33:03 restricted M1_124–134_ and HLA-B*44:03 restricted NP_319–330_, presentations were detected at 0.5 h after infection (beginning after the first-hour infection), however, at that time the presentation of M1_58–66_ was barely detectable. M1_124–134_ presentation rapidly activated 50% of its antigen-specific T cells approximately 2 h after rVV-M1 infection (Figure 5c), whereas M1_58–66_ reached the same level after an additional 2 h and NP_319–330_ after 4 h (Figure 5a). HLA-B*44:03 restricted epitopes, M2_7–15_, NS1_195–203_, PB2_39–49_ and HA_445–453_, showed similar slower presentation kinetics with detectable presentation at 2 h and 50% presentation at approximately 6 h after infection (Figure 5a). The presentation of the HLA-A*24:02-restricted epitopes, PA_130–138_ and PB1_498–505_, were faster as they were detected at 2 h and reached 50% at 4 h (Figure 5b). The presentation of HLA-A*33:03-restricted epitope, M1_124–134_ was faster than that of M1_58–66_. M1_124–134_ became detectable at 0.5 h after infection and reached 50% saturating presentation at 2 h (as previously mentioned), however, NA_337–346_ presentation was slightly slower. Taken together, these results indicate the speed of antigen presentation did not seem to correlate with the magnitude of CD8^+^ T cell response to each epitope albeit these responses were all assessed after in vitro T cell expansion using rVV-infected APCs.

## 4. Discussion

In this study, we report a broad-based CD8^+^ T cell response, rather than one with a typical immunodominance hierarchy, in an IAV-experienced healthy individual. Combining a systematic antigen screen approach with epitope prediction, CD8^+^ T cell responses to nine IAV proteins (M1, M2, NP, NS1, PA, PB1, PB2, HA and NA) except for NS2 and PB1-F2 and their minimal epitope sequences were identified. These epitopes are restricted to HLA-B*44:03, HLA-A*24:02 and HLA-A*33:03 and seven out of the nine epitopes are novel (NP_319–330_, M1_124–134_, M2_7–15_, NA_337–346_, PB2_39–49_, HA_445–453_ and NS1_195–203_). Most of these novel epitopes are highly conserved among strains of H1N1 and H3N2 that circulated in Australia and other parts of the world.

Several HLA molecules such as HLA-A*02:01, HLA-A*03:01, HLA-B*57:01, HLA-B*18:01 and HLA-B*08:01 present IAV peptides to stimulate robust CD8^+^ T cell responses against IAV infection, whereas HLA-A*24:02, HLA-A*01:01, HLA-A*68:01, and HLA-B*15:01 often stimulate more limited CD8^+^ T cell responses specific to IAV [38]. In addition, HLA-A*24:02 was reported to be associated with severe pH1N1 disease and is an allele that is highly expressed in Indigenous Australians and Alaskans [39,40]. Due to the lack of immunodominant epitope presentation by this HLA, especially in the absence of other HLAs that present immunodominant epitopes, such as HLA-A*02:01, HLA-B*08:01, HLA-B*15:01, and HLA-B*35:03 [5,6,21,41], may indicate HLA-A*24:02 does not present an immunodominant IAV epitope and therefore plays a limited role during an anti-IAV cellular immune response which would result in severe pH1N1 disease [42]. Additionally, Quinones-Parra et al. [38] investigated pre-existing memory CD8^+^ T cell pools specific for the conserved H7N9 epitopes and found eight conserved NP epitopes including an HLA-A*24:02-restricted epitope NP_39–47_ (A*24:02-NP_39_). They then showed CD8^+^ T cell responses to six of the eight conserved epitopes, however, no CD8^+^ T cell specific for A*24:02-NP_39_ was detected ex vivo, suggesting although NP_39–47_ is a conserved epitope, it is a weak one that does not elicit a significant CD8^+^ T cell memory response. Such CD8^+^ T cell responses are believed to be less protective. Our results might at least partially explain why Indigenous Australians were more susceptible to severe IAV infection and had higher mortality rates than other Australian populations during the 2009 H1N1 pandemic [43,44]. To date, researchers have mainly focused on assessing immunogenic epitopes for common HLA molecules, such as HLA-A*02:01, with much less attention paid to the HLA molecules associated with Indigenous Australians.

Immunodominance has been studied for decades. In many cases, removing the most dominant T cell epitope enables the T cells specific to the subdominant epitopes to expand more completely [45,46]. Although the exact mechanisms associated with this phenomenon remain poorly understood, several factors have been shown to influence it, such as peptide-binding ability, T cell competition for particular pMHC-I on the APC surface, dominant T cells actively suppressing other T cell development, as well as other factors [46]. For example, Kedl et al. generated CD8^+^ T cells to dominant epitope SIINFEKL (ova8) and subdominant epitope KRVVFDKL from Ovalbumin in H-2^b^ mice; when high-affinity TCR transgenic T cells (OT1 T cells) for ova8 were transferred, they completely inhibited the response of host OVA-specific T cell response to either epitope due to APC level competition [47]. However, in our case, no functional immunodominant CD8^+^ T cell responses were discovered, indicating that other mechanisms rather than APC level competition are more likely.

Early evidence demonstrated that T cell precursor frequency and the efficiency of peptide generation could influence immunodominance hierarchy [48]. For example, Day et al. found that co-expression of H-2K^k^ and H-2D^b^ greatly diminished CD8^+^ T cell response to the H-2D^b^-restricted immunodominant epitope PA_224–233_, as part of the T cell repertoire was deleted by having H-2K^k^, while T cell responses to H-2D^b^ restricted NP_366–374_ remained unaltered [49]. However, more recently a study from Neller et al. demonstrated that T cell precursor frequency might not correlate closely with the immunodominance hierarchy after being pathogen challenged [50]. It is possible none of the epitopes we identified in this study could stimulate a typical immunodominant response.

Even though MHC restriction should be the most significant determining factor, it is often under-considered and sometimes not considered, especially in syngeneic murine systems in which MHC are pre-fixed. In humans, since HLA molecules are extremely polymorphic, the overall immunodominance hierarchy is shaped up by the CD8^+^ T cell responses restricted to all the involved HLA alleles expressed by the individual. For example, HLA-A*02:01-M1_58–66_ complexes stimulate an immunodominant CD8^+^ T cell response in many HLA-A*02:01 donors, but not in those who also express HLA-A*01:01, HLA-B*08:01, HLA-B*15:01, HLA-B*18:01, and HLA-B*35:03 [5]. Additionally, some epitopes can be restricted to a few different HLA alleles. For example, M1_125–134_ has been reported as an HLA-A*33:01 epitope [33]. Interestingly, in this study, we show this same epitope can be restricted to HLA-A*33:03, therefore, it is envisaged if one individual expressed both HLA-A*33:03 and HLA-A*33:01, the response to M1_125–134_ might be boosted into an immunodominant status.

HLA combinations were shown to be one of the most important elements contributing to CD8^+^ T cell immunodominance hierarchy [35]. Akram et al. demonstrated during IAV infection in HLA transgenic mice that only certain combinations of HLA allele co-expression enabled immunodominant CD8^+^ T cell responses. For instance, the IAV immunodominant CD8^+^ T cell response could only be detected in IAV-infected HLA-B*7/HLA-B*27 transgenic mice but not in HLA-A*2/ HLA-B*7 or HLA-A*2/HLA-B*27 mice [51]. Moreover, Boon et al. reported HLA alleles were not used equally and there was a hierarchy that existed between HLA alleles; for example, HLA-B*27:05 and HLA-B*35:01 were preferentially used in IAV-specific CD8^+^ T cell response, and HLA-B*08:01 and HLA-A*01:01 seemed to contribute less [17]. According to the above results, co-expression of certain alleles may determine not only which HLA may present an immunodominant CD8^+^ T cell response but also whether there is immunodominant CD8^+^ T cell response at all. In HIV studies, epitopes restricted by HLA-B molecules were significantly more frequently recognised than those presented by HLA-A or HLA-C molecules [52]. Interestingly in our case, the study subject expresses HLA-A*24:02, HLA-A*33:03, HLA-B*44:03, HLA-B*46:01, HLA-Cw*01:02, and HLA-Cw*07:01, yet the HLA-B*46:01 and the HLA-C alleles contributed little to the overall anti-IAV response. Furthermore, this set of HLA was not able to support an immunodominant CD8^+^ T cell response during IAV infection.

It is also worth considering infection exposure and vaccination history, as a specific viral strain might affect the outcome of CD8^+^ T cell response and the immunodominance hierarchy. Although little is known about the precise IAV exposure history of this donor, we have compared the conservancy of novel epitopes among all the IAV strains circulated in Australia in the past 50 years and found two out of the seven identified novel epitopes are 100% conserved among these strains (Table 2). These results suggest that our assessment is likely valid even though only one IAV strain (PR8) was used for conducting the experiments in this study.

Chronic virus infections, such as HIV and HCMV are characterised by the accumulation of antigen-specific CD8^+^ T cells specific for immunodominant epitopes [53]. However, IAV usually causes acute infection which might not allow CD8^+^ T cells with a particular specificity to accumulate significantly. Therefore, if one does not possess a set of HLAs that is able to elicit a robust IAV-specific CD8^+^ T cell response, such individual may have a higher risk for developing severe IAV-related diseases. Our present results provide better understanding of IAV-specific CD8^+^ T cell responses in humans, especially in high-risk groups with HLAs that do not present major CD8^+^ T cell epitopes.

In this study, the novel peptides discovered are presented by HLA-B*44:03 and HLA-A*33:03. According to the IEDB, there is no HLA-B*44:03 restricted IAV immunodominant CD8^+^ T cell epitope published previously; moreover, no HLA-A*33:03 restricted IAV epitope has been identified so far. It is possible the set of HLA alleles this individual expresses failed to present any immunodominant epitope resulting in a broad-based, relatively low-magnitude CD8^+^ T cell response during IAV infection. Although we have only studied one such case on the broad-based CD8^+^ T cell responses without the typical immunodominance hierarchy, we believe this phenomenon might be wider spread than previously expected as published studies often only focused on a relatively small numbers of high frequency HLA alleles. It is possible many other HLA alleles, just like the alleles HLA-A*33:03 and HLA-B*44:03 investigated in this study, might not present IAV epitopes that are capable of stimulating immunodominant response partly because their peptide motifs are too stringent. As both the HLA haplotypes encoding HLA-A*24:02—HLA-B*44:03 and the ones encoding HLA- A*33:03—HLA-B*44:03 are very rare in Caucasian populations, quite rare in African and indigenous populations and generally less than 1% in most Asian groups except for China Jingpo minority and South Koreans (among these populations HLA-A*33:03—HLA-B*44:03 associated haplotypes can reach 5% according to http://www.allelefrequencies.net/, accessed on 25 April 2021), we have not been able to confirm our observation in another donor. It would be interesting and worthwhile to either confirm our observation using donor samples collected from the above mentioned Jingpo minority and/or South Korean populations or to carry out a large-scale screen to identify the HLAs associated with such responses in the future. Alternatively, it might be possible to HLA-type all the IAV-infected severe individuals to see whether there is any special association between their HLA alleles and their disease severity. Such knowledge may allow us to predict high-risk individuals during IAV infection.

## Figures and Tables

**Figure 1 viruses-13-01080-f001:**
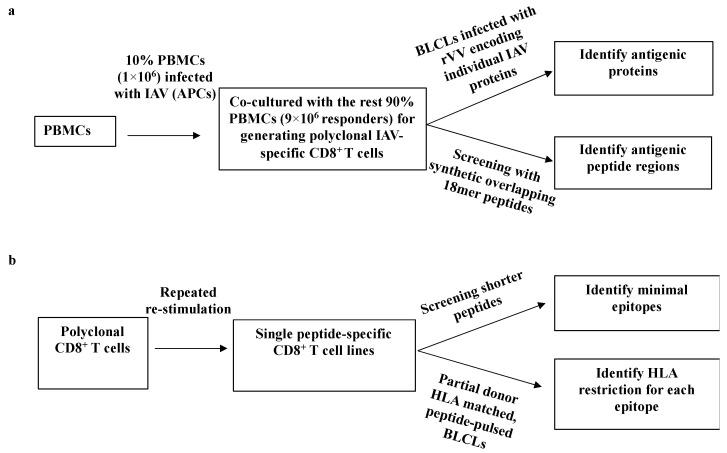
A systematic approach for identifying IAV-specific CD8^+^ T cell responses. (**a**) Establishing polyclonal IAV-specific CD8^+^ T cells. After two weeks culture, CD8^+^ T cells were screened by a panel of individual IAV proteins encoded by rVV to identify specific antigenic proteins. After knowing the specific antigenic protein, CD8^+^ T cells were screened with synthetic overlapping 18mer peptides to determine the specific antigenic peptide regions within the protein. (**b**) Generating individual peptide-specific CD8^+^ T cell lines using individual peptide-pulsed APCs to expand IAV-specific CD8^+^ T cells enriched via CD3CD8 downregulation-guided sorting. Two weeks later, CD8^+^ T cells were tested with shorter peptides to assess the core sequence and HLA restriction of each particular epitope.

**Figure 2 viruses-13-01080-f002:**
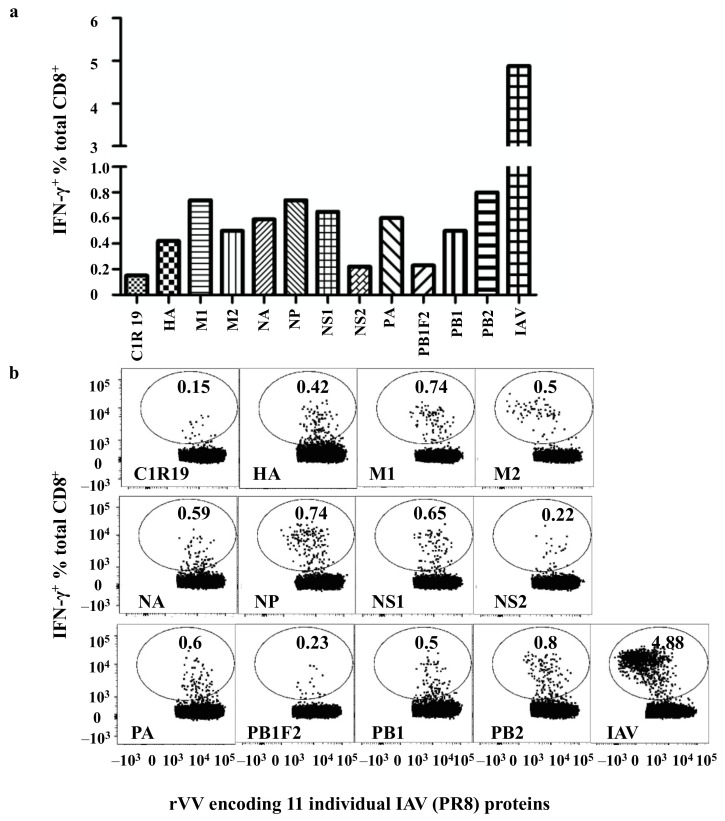
Broad-based CD8^+^ T cell responses to multiple IAV proteins. (**a**) Autologous BLCLs were infected with rVVs encoding individual IAV proteins for 16 hours before being used as APCs to assess CD8^+^ T cell responses via a standard ICS for IFN-γ production. (**b**) FACS profiles from this rVV screen.

**Figure 3 viruses-13-01080-f003:**
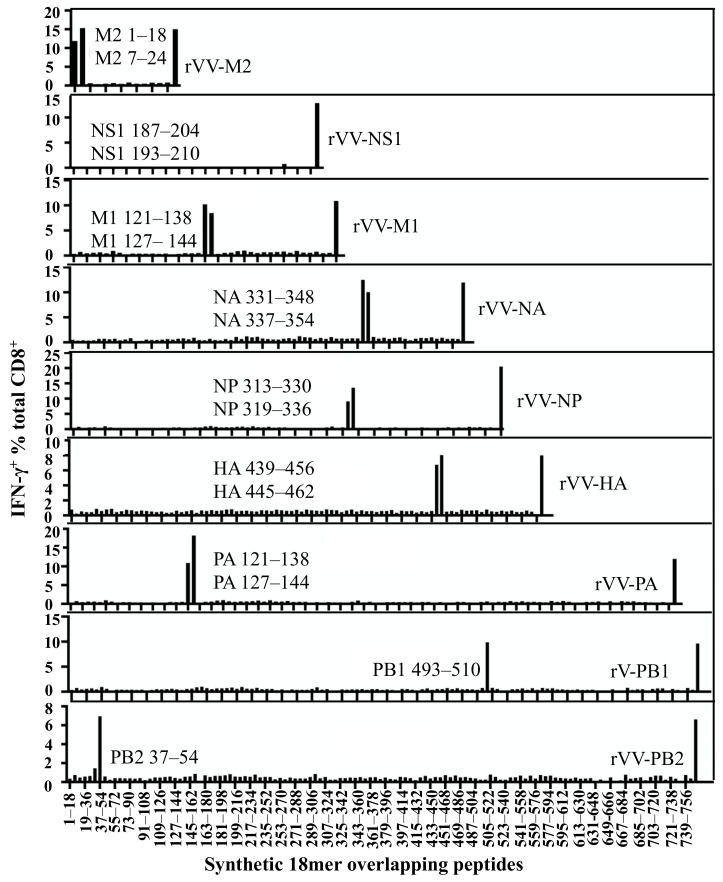
Identification of specific antigenic regions using synthetic 18mer overlapping peptides. The CD8^+^ T cell responses of individual IAV proteins were assessed with 18mer overlapping peptides covering the full sequences of M2, M1, NP, NS1, PB1, PB2, HA, NA, and PA in a standard ICS assay. In the same experiment, APCs infected by each rVV encoding individual IAV proteins were used as positive controls shown at the far right in each graph.

**Figure 4 viruses-13-01080-f004:**
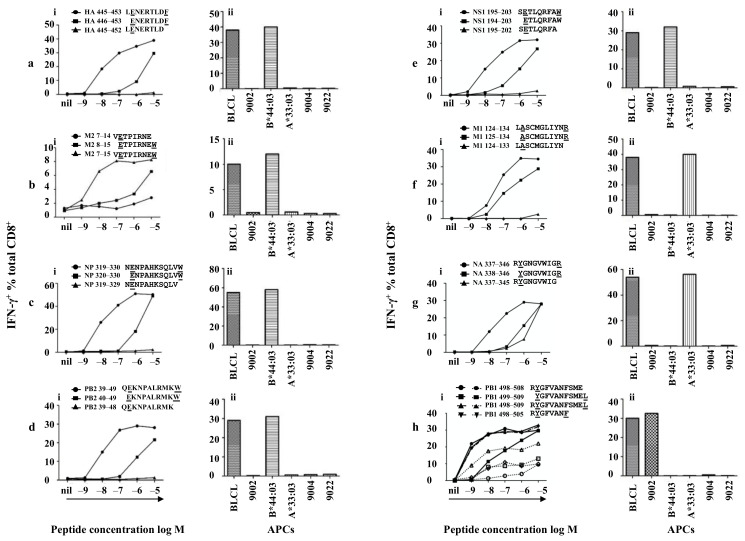
Minimal epitopes of differing lengths are presented by HLA-B*44:03, HLA-A*33:03 and HLA-A*24:02. Peptide-specific CD8^+^ T cells were established as described in the legend to Figure 1b (**a**–**h**) various CD8^+^ T cell specificities were assessed via standard ICS for the production of IFN-γ with different peptides and APCs. (**i**) peptide titrations of the minimal peptides and their truncated peptides, (**ii**) HLA restrictions of each minimal epitope. (**a**–**g**) peptide titrations conducted in the absence of FCS, (**h**) PB1 peptide titrations conducted in the presence of FCS (shown in solid lines) or absence of FCS (shown in dotted lines).

**Figure 5 viruses-13-01080-f005:**
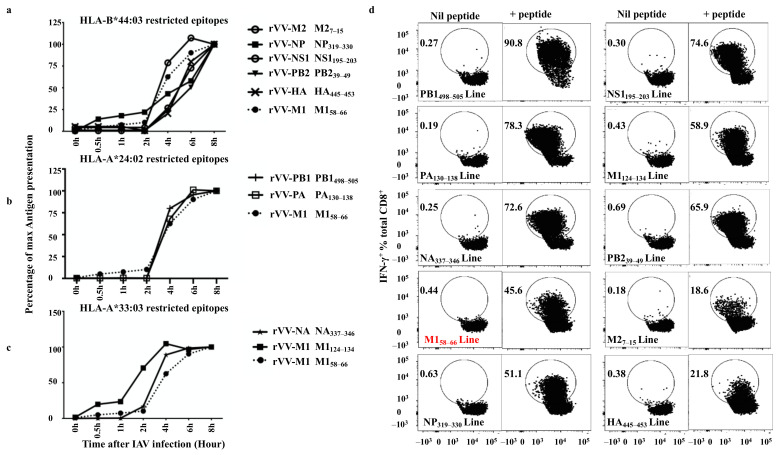
Most novel CD8^+^ T cell epitopes are presented with a fast kinetics. CD8^+^ T cells were activated by rVV-infected BLCLs at various time points (0, 0.5, 2, 4, 6, and 8 h) after the first-hour infection, then BFA and T cells were added to “freeze” and read out antigen presentation at that time point after rVV infection. The antigen presentation, reflected by the amount of epitope-specific T cell activation was assessed for IFN-γ production by ICS as described in the method. (**a**) HLA-B*44:03 restricted epitope presentation is compared to that of a well-studied immunodominant epitope M1_58–66_ (shown in dotted line). (**b**) HLA-A*24:02 restricted epitope presentation is compared to that of M1_58–66_ (shown in dotted line). (**c**) HLA-A*33:03 restricted epitope presentation is compared to that of M1_58–66_ (shown in dotted line). This experiment was performed and analysed together. For clarity, the results are displayed according to single HLA restriction of the epitopes. Data shown are representative of two independent experiments with similar results. (**d**) The FACS profiles showing the purities of the CD8^+^ T cell lines specific for PB1_498–505_, NS1_195–203_, PA_130–138_, M1_124–134_, NA_337–346_, PB2_39–49_, M1_58–66_, M2_7–15_, NP_319–330_, and HA_445–453_ used in (**a**–**c**). Nil peptide: APC only control; +peptide: APCs were pulsed with each individual peptide at 10^−6^ M for 1 h in RF-10 before being incubated with antigen-specific CD8^+^ T cells and assessed in a standard ICS.

**Table 1 viruses-13-01080-t001:** HLA information of APCs.

APCs	HLA-A	HLA-B	HLA-Cw
auto-BLCL	**24:02**, **33:03**	**44:03**, **46:01**	**01:02**, **07:01**
9002	**24:02**	14:02	02:02, 08:02
C1R-B*44:03		**44:03**, residual 35:03	04:01
C1R-A*33:03	**33:03**	residual 35:03	04:01
9004	02:01	27:05	**01:02**
9022	01:01	08:01	**07:01**

Note: HLA alleles matching Donor’s are shown in bold.

**Table 2 viruses-13-01080-t002:** Conservancy of the novel epitope sequences among all the IAV strains circulated in Australia during 1971–2021.

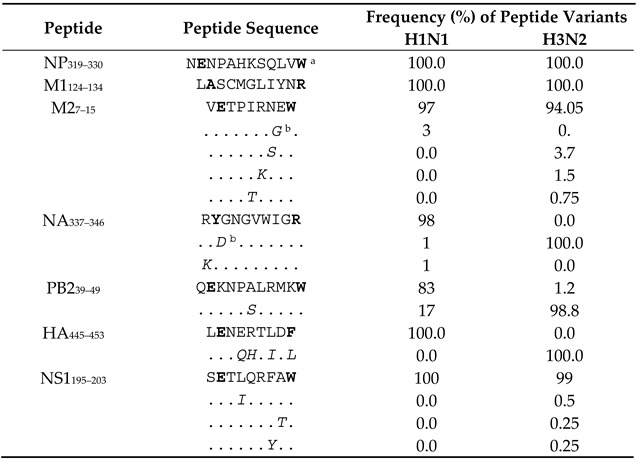

Note: Australian circulated H1N1 and H3N2 IAV strain full-length sequences for the studied proteins were downloaded from the NCBI influenza resource database (https://www.ncbi.nlm.nih.gov/genomes/FLU/Database/nph-select.cgi?go=database, accessed on 21 April 2021). Specific searching criteria for downloading were country/region Australia), protein (NP/M1/M2/NA/PB2/HA/NS1), subtype (H1N1/H3N2), sequence length (full-length only). Collected data from 1971 to 2021 show identical sequences by the oldest sequence in the group. ^a^ HLA anchor residues are shown in bold; ^b^ Variations from the PR8 strain sequences are shown in italics.

**Table 3 viruses-13-01080-t003:** All the IAV strains used for epitope conservation analysis.

H1N1	H3N2		
A/Cairns North/INS597/2010	A/Australia/18/2009	A/Perth/1055/2014	A/Sydney/852M/2002
A/Sydney/DD3-02/2009	A/Australia/NHRC0001/2003	A/Perth/113/2015	A/Sydney/329Q/2003
A/Victoria/2005/2009	A/Australia/NHRC0001/2005	A/Perth/153/2012	A/Sydney/419J/2004
A/Brisbane/59/2007	A/Brisbane/9/2006	A/Perth/16/2009	A/Sydney/348N/2006
A/New South Wales/18/1999(H1N1)	A/Brisbane/1/2012	A/Perth/164/2013	A/Sydney/257U/2009
A/South Australia/45/2000	A/Brisbane/1/2013	A/Perth/22/2016	A/Sydney/DD2-01/2010
A/Western Australia/18/2001	A/Brisbane/1/2017	A/Queensland/1/2000	A/Tasmania/60/2012
A/Brisbane/193/2004	A/Brisbane/10/2007	A/Queensland/11/2001	A/Tasmania/11/2014
A/South Australia/55/2005	A/Brisbane/100/2014	A/Queensland/21/2002	A/Tasmania/1005/2015
A/Western Australia/77/2005	A/Brisbane/1000/2015	A/Queensland/31/2003	A/Tasmania/1001/2016
A/Victoria/500/2006	A/Brisbane/1004/2016	A/Queensland/41/2004	A/Townsville/1004/2012
A/Brisbane/297/2006	A/Brisbane/11/2010	A/Queensland/51/2005	A/Townsville/10/2015
A/Perth/1/2007	A/Canberra/1001/2012	A/South Australia/1/2012	A/Townsville/1003/2016
A/Victoria/501/2007	A/Canberra/1001/2016	A/South Australia/1/2015	A/Victoria/75/1995
A/South Australia/423/2007	A/Canberra/113/2013	A/South Australia/1002/2014	A/Victoria/210/2009
A/Sydney/581/2007	A/Canberra/12/2015	A/South Australia/1007/2016	A/Victoria/361/2011
A/South Australia/2001/2009	A/Canberra/2/2014	A/South Australia/15/1994	A/Victoria/1002/2012
A/Brisbane/17/2009	A/Darwin/1000/2015	A/South Australia/15/2000	A/Victoria/504/2013
A/Australia/1/2009	A/Darwin/1001/2012	A/South Australia/18/2005	A/Victoria/1012/2014
A/swine/VIC/09-02767-01/2009	A/Darwin/1002/2016	A/South Australia/2/2013	A/Victoria/1000/2015
A/swine/QLD/09-02865-07/2009	A/Darwin/4/2005	A/Sydney/223G/2005	A/Victoria/109/2016
A/Sydney/DD3-37/2010	A/New South Wales/13/1999	A/Sydney/0017/2007	A/Western Australia/1/2000
A/North Fitzroy/INS463/2010	A/New South Wales/27/2000	A/Sydney/1001/2012	A/Western Australia/13/2001
A/Melbourne/INS467/2010	A/Newcastle/1/2013	A/Sydney/1014/2013	A/Western Australia/23/2002
A/Westmead/INS524/2010	A/Newcastle/1000/2016	A/Sydney/1004/2014	A/Western Australia/37/2003
A/Sydney/DD3-58/2011	A/Newcastle/1001/2015	A/Sydney/10/2015	A/Western Australia/51/2004
A/Westmead/INS3_659/2011	A/Newcastle/1016/2014	A/Sydney/101/2016	A/Western Australia/65/2005
A/Darlinghurst/INS3_645/2011	A/PERTH/1/2003	A/Sydney/5/1997	
A/Melbourne/INS3_670/2011	A/Perth/10/2010	A/Sydney/405A/2001	

Note: This table only includes part of the IAV strains circulated in Australia during the 1971–2021 period.

## Data Availability

Not applicable.

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
