# Peer review of "Broad-Based Influenza-Specific CD8+ T Cell Response without the Typical Immunodominance Hierarchy and Its Potential Implication"

_viruses, 2021, doi:10.3390/v13061080_

Round 1
Reviewer 1 Report
This study performs a screen to determine the breadth of the IAV-specific CD8 T cell response in individuals. It also investigates rate of antigen presentation as a potential correlate to immunodominance hierarchy and performs alignments to yield insights into potential conservation of CD8 T cell epitopes between previously circulating IAV strains.
However, the data presented to support the conclusions of the study are limited and there is much speculation in place of experimental evidence. Below are a few specific comments highlighting the lack of data and preliminary nature of the study presented as is:
- For the initial screen, the number of individuals assessed the results related to the CD8 T cell repertoire are not readily apparent.
- A major limitation of the study is that after the initial screen, the study investigates epitope spreading only in one individual of those initially assessed for CD8 T cell responses. It is unclear how representative this type of response is in a larger population.
- There are no functional studies to assess potential conserved responses to IAV epitopes identified, only alignments highlighting anchor residues.
- The study concludes that the speed of antigen presentation did not correlate with the magnitude of the CD8 T cell response. Given the stated caveat that responses were assessed after T cell expansion, this does not seem relevant.
Reviewer 2 Report
Title of the Manuscript: Broad-based Influenza-specific CD8+ T cell response without typical immunodominance hierarchy and its potential implication
Manuscript ID: viruses-1222310
Summary of the manuscript:
Huang et al., assessed CD8+ T cell responses to IAV proteins using rVVs encoded by IAV proteins and PBMC’s of a healthy donor. Huang et al., have identified 9 proteins along with their minimal epitope sequences that are restricted to HLA-B*44:03, HLA- 21 A*24:02 and HLA-A*33:03 alleles. Seven out of these nine epitopes were novel and are highly conserved among H1N1 and H3N2 strains that circulated in Australia.
Huang et al., reports interesting findings
- Broad CD8+ T cell responses to IAV proteins
- Responses were mostly sub-dominant without typical immunodominance hierarchy
Overall, results from this study provides additional insights about new CD8+ T cell epitopes of influenza A virus (IAV), might allow for the development of T cell-inducing vaccines that provide protection across different strains and subtypes. All though these results are needed to be validated further by making tetramers and see if these can generate responsesin-vivo.
Comments:
- Since the entire manuscript is based on IFN-g responses, I recommend authors to provide representative flow plots for CD8 + T cells IFN-g staining/ICS in supplementary
Round 2
Reviewer 1 Report
The authors have addressed all reviewer concerns.